# Emotional Arousal Impacts Physical Health in Dogs: A Review of Factors Influencing Arousal, with Exemplary Case and Framework

**DOI:** 10.3390/ani13030465

**Published:** 2023-01-28

**Authors:** Carrie Tooley, Sarah E. Heath

**Affiliations:** Behavioural Referrals Veterinary Practice, Chester CH2 1RE, UK

**Keywords:** emotion, arousal, behaviour, behavioural medicine, stress, fly catching, neurological disorder, dog

## Abstract

**Simple Summary:**

Emotional arousal can impact physiological health in both human-animals and non-human animals. Currently, most work in the field of veterinary behavioural medicine focuses on reducing the activation of the protective emotional systems (including, but not limited to fear-anxiety). The assessment and treatment of all emotional systems, with equal weight placed on the engaging systems and protective systems, has not traditionally been considered in the treatment of physiological health of veterinary species. This article discusses the relationship between emotional arousal and physiological health with particular reference to the role of sleep. A case report of a seven month old male entire Cocker Spaniel showing fly-snapping behaviour is presented. The emotional health assessment and treatment of this case is described to demonstrate the effect of an appropriate behavioural medicine treatment plan in cases such as this. The authors put forward the argument that an emotional health assessment should be considered an essential component of the work up of all such cases.

**Abstract:**

Excessive emotional arousal has been shown to impact physiological health in both veterinary species and human animals. The focus of work in many models of veterinary behavioural medicine has predominantly been associated with reducing activation of the protective emotional systems; in particular, fear-anxiety. The management of the engaging emotional systems of desire-seeking, social play, care and lust has not traditionally been considered in the treatment of physiological health of veterinary species. This article reviews the literature in both veterinary and human fields on the relationship between emotional arousal of both protective and engaging emotional systems and physical health conditions. The current literature describing the regulatory control of sleep on emotional arousal is also discussed. An exemplary case report of a seven month old male entire Cocker Spaniel showing fly-snapping behaviour which had been non-responsive to leviteracetam (Keppra) is presented. The emotional health assessment and treatment of this case is described along with the short and long term (fourteen month follow up) outcomes to demonstrate that some patients presenting in this way can be effectively managed with an appropriate behavioural medicine treatment plan. The authors put forward the argument that an emotional health assessment should be considered an essential component of the work up of all such cases.

## 1. Introduction

The interplay between emotional health and physical health is widely reported [1,2,3,4,5,6,7,8,9,10,11,12]. Despite this, it remains uncommon for emotional health to be formally assessed in veterinary patients presenting with chronic physiological conditions. In an excellent recent paper by Packer et al. [13] the link between epilepsy and emotional arousal in human and canine patients is explored and the conclusion drawn that veterinary behavioural medicine intervention is indicated in the management of epilepsy cases. That paper focuses on the treatment of protective emotional responses described as “stress” and anxiety within the behavioural medicine approach. It does not consider the influence of other protective emotional responses or the engaging emotional responses [14]. The Heath Model of Emotional Health [15,16] leads to a clinical approach to veterinary behavioural medicine which has a strong focus on assessing total (protecting plus engaging) emotional arousal and ensuring both are appropriate to achieve optimal emotional health. This article reviews current literature on emotional arousal, the factors that influence excessive arousal and processes which regulate emotional arousal. The signs of excessive emotional arousal are discussed and a case example presented, with a framework described for approaching such cases.

## 2. Emotional Arousal

The most commonly accepted paradigm for the study of emotions in animals is Panksepp’s affective systems [14]. Panksepp outlines four positive and three negative affective systems which are the emotional motivations influencing behaviours. Positive systems motivate an individual to engage with a stimulus and negative systems motivate an individual to protect themselves from a stimulus. The positive systems are desire-seeking (an appetite for resource acquisition which motivates a range of behaviours including object play, scent work, predation and the sourcing of vital resources), social-play (positive interactions with conspecifics such as rough and tumble play), care (predominantly motivating nurture of offspring) and lust (reproductive motivation promoting courtship and mating). The negative systems in Panksepp’s paradigm are fear-anxiety (protective response to possible and true threats), rage (emotion activated by the thwarting of expectation or achievement), panic-grief (activated by the loss of proximity to a nurturing caregiver) and, in latter versions of the paradigm, pain (in earlier versions this was part of the fear-anxiety-pain protective system). Many individuals working in the field of veterinary behavioural medicine use this paradigm with some changes to terminology. For example, it is typical for “rage” to be replaced by “frustration” [17,18,19,20,21] to aid caregiver understanding. In addition, fear and anxiety are often discussed in terms of sharing the same neuroanatomy and neurotransmitters, but being separate from a functional point of view [18,20,22]. Fear is recognised as active when a perceived threat is present [20] and anxiety recognised as active in potentially threatening situations (such as novel or unpredictable environments or interactions) [20,23,24]. The distinction between these systems is clinically important as they are active in different contexts and should be treated as such. In addition, the emotion of anxiety can be situational (due to uncertainty or anticipation of an unwanted outcome in a specific context, such as veterinary examinations) or generalised (triggered in a wide variety of contexts). The Heath Model [15,16] uses the terminology of “engaging and protective” in place of Panksepp’s “positive and negative” emotional systems. The term “protective” is used to describe the function of the emotions of fear, anxiety, frustration, panic-grief and pain and “engaging” to describe the function of the emotions of desire-seeking, social play, care and lust. This terminology was developed in response to clinical experience and reported confusion on the part of caregivers who perceived the term “negative” to be synonymous with “bad” or “detrimental”. This in turn led them to consider the presence of these emotions to always be problematic and to think that optimal emotional health could only be restored if these emotions were removed. By using the terms protective and engaging the emphasis within the Heath Model is that all of the emotional systems are healthy and beneficial to the individual, so long as they are experienced in proportion to the environment the individual is in and result in are healthy, promoting appropriate behavioural responses which are able to succeed. The aim is for caregivers to more accurately and easily comprehend the biological function of these emotions and the motivations for their pet’s behaviour. In the case of the engaging (positive) emotions these behavioural responses have the purpose of connecting the individual to something or someone who is of benefit for their survival and the protective (negative) emotions are associated with behaviours which protect the individual from real or potential, physical or emotional harm. For example, the action of running out of the way of a moving lorry, thereby preserving life, is motivated by the fear emotional system and is of undoubted benefit to the individual.

Many of the common “problem” behaviours presenting in a clinical context are primarily motivated by protective emotions and this is often the main focus of behavioural consultations. When assessing emotional health as a contributing factor to a physiological condition it is important to consider the total emotional arousal present; both protective and engaging. If total emotional arousal is excessive in duration or intensity an individual is likely to show physiological and/or behavioural compromise. The Heath Model uses the emotional sink analogy to discuss this with caregivers with the hot tap representing protective emotional activation and the cold tap representing engaging emotional activation. It is the water from both of those taps which fill the same “emotional sink” and can result in an individual becoming emotionally overwhelmed [15,16].

## 3. Temperament and Traits

Temperament can be difficult to assess objectively and there are a multitude of labels in the literature with variable interpretations [25]. Formal assessment of temperamental traits is possible via validated questionnaires such as the PANAS (Positive and Negative Activation Scale) [26,27] and DIAS (Dog Impulsivity Assessment Scale) [28] which facilitate numerical assessment of emotional bias and impulsivity in relation to a normal range. Tracking of progress via pre and post intervention use of the questionnaires can provide valuable progress data for clinicians. In addition, assessment via these tools in a large cohort of similar patients would provide useful information for future research into the aetiology and treatment of certain presenting problems.

Much is discussed regarding the impact of breed on temperament and certain breeds are portrayed in the media to show higher levels of certain traits [29]. There is some evidence to support breed characteristics [30], especially when considering breed groups such as “Herding dogs, Hounds, Working dogs, Toy dogs and Non-sporting dogs” [31] rather than specific individual breeds. However, the scientific study of this is very challenging [32] and conclusions should be drawn with caution. Some research suggests that there is more within-breed than between-breed variation in behavioural expression [33] and that this is particularly evident when considering working compared to show lines within a breed [34]. This is supported by the authors’ experience, seeing individual patients where working line individuals of a variety of breeds are diagnosed with higher levels of generalised anxiety. Generalised anxiety can often be functional in working dogs where the presence of the resulting vigilance to both environment and handler can improve performance. The expression of fear and anxiety is a heritable trait and has been shown to be inadvertently selected for by breeders in some working dogs [22], increasing prevalence.

## 4. The Role of Sleep

One important factor in emotional and physiological health for any individual is sleep [35,36]. The relationship between sleep and emotional health is complex [37] and sleep research in veterinary species in is its infancy. Sleep, in addition to licking and chewing behaviours, are referred to as drainage behaviours in the Heath Model. An appropriate balance between the input of emotion through the metaphorical taps, the size of the sink, representing the emotional capacity of the individual, and the drainage of emotion out through the metaphorical outlet pipe, is essential if an individual is to maintain a level of emotional arousal, represented by the amount of water in the sink, which remains within their emotional capacity and therefore be emotionally healthy. In the authors’ clinical experience, the role of these drainage behaviours in reducing emotional arousal is essential to patient welfare. In addition to the effects on emotional health, sleep has been shown to influence cognitive health via effects on memory consolidation [38,39,40].

A “healthy” canine sleep duration is difficult to determine with a variety of reported periods in the literature [41,42,43]. A recent study suggests canine individuals with low sleep duration whilst their caregivers are out of bed, tend to be individuals who are easily disturbed (by external environmental stimuli or in-home social stimuli) from sleep. The more frequent these disturbances, the lower the sleep duration and the worse the severity of “problem” behaviours those individuals are reported, by their caregivers, to demonstrate [44]. The main causal factor for this is not established. Are individuals with more protectively biased emotional states, already more likely to show problem behaviours, achieving only vigilant states of sleep due to the perception of risk in the environment [45]? Are individuals with an engaging emotional bias being woken from deep (non-vigilant) sleep by significant stimuli (such as nearby fireworks or a household baby crying) and therefore sleep deprived and showing problem behaviours as a result?

Sleep duration is important for emotional health, but it is not the only factor. The type of sleep achieved is highly significant. Rapid eye movement (REM) sleep is thought to modulate emotional arousal in human-animals, a process by which an individual achieves an increased capacity to cope with total emotional arousal in the waking phase after sufficient REM sleep [43,46,47,48]. In contrast, non-REM sleep is thought to influence cognitive learning in both human-animals and dogs [49]. Human-animals with reduced REM sleep show increased anxiety [50], impulsivity [51] and protective emotional intensity [52] in response to aversive stimuli. In addition, REM sleep in human-animals is thought to facilitate memory processing; the filtering of experiences such that useful memories are retained for future use, without the full experience being re-lived during future recollection [53,54,55,56].

In the authors’ experience, caregivers often struggle to accurately identify the duration of sleep, or even rest, that their pets achieve. Time budgets are a useful way to elicit what activities a patient is undertaking, and for what duration, throughout their day. They have been used extensively to report dog behaviour in the literature in a wide variety of contexts from free roaming [57,58] to laboratory [59] dogs.

## 5. Clinical Signs of Excessive or Inappropriate Emotional Arousal

The interplay between emotional health, physical health and expression of problem behaviours is complex. Fatjó and Bowen [4] state “the border between medically and non-medically related behaviour problems is becoming less well defined as more information is available”. Excessive emotional arousal can directly motivate certain behaviours and also have acute and chronic impacts on physiological health.

Acute emotional arousal promotes activation of the sympatho-adreno-medullary (SAM) axis, promoting catecholamine release and the hypothalamic-pituitary-adrenal (HPA) axis promoting glucocorticoid release [60]. Additionally, the sympathetic nervous system (SNS) increases heart rate, blood pressure, promotes glycogenolysis and ceases gastrointestinal peristalsis [61]. These processes are functional if an individual needs to deal with a genuine threat in their environment, but can lead to physiological problems if this is not the case. For example, a cat consuming a meal in sight of another cat will, in the authors experience, typically show behavioural signs of protective emotional bias; ears either flattened or orientated in vigilance to the observing cat, tension in the facial and body musculature, abnormally rapid speed of eating and absence of the typical relaxed crouched eating posture. Often cats displaying this behaviour are experiencing, at least, the protective emotions of frustration and anxiety along with the engaging emotion of desire-seeking. This intense emotional arousal will activate the SAM and HPA axes, along with SNS activation, all of which are counter-productive for digestion.

Chronic, excessive protective emotional arousal causes prolonged, excessive corticosteroid production via the Hypothylamic Pituitary Adrenal (HPA) axis [20]. The effects of chronic excessive engaging emotional arousal on the HPA axis have not been studied or established in veterinary fields, though the basic processing which generates emotional arousal shares similar properties between the two valences [20]. Cortisol has extremely diverse effects in the body [62] and as a result, chronically raised cortisol effects a number of physiological conditions. The impact of emotional arousal is widely reported in feline lower urinary tract disease [1,2] and pain [3], is known to precede neurological episodes such as movement disorders in people [6,11] and veterinary species [3,5,8,9,10], is thought to predispose an individual to dermatological conditions [12] and down-regulate immune function [60].

Excessive emotional arousal can directly and acutely motivate certain behaviours. Excessive arousal of each affective system will promote increased intensity of the subsequent behavioural expression [4]. Increasing towards or decreasing from intense emotional arousal of any of the affective systems correlates with performance of species-specific displacement behaviours which are represented in the Heath Model sink analogy by the overflow hole at the top of the sink.

Excessive emotional arousal is associated with cases where individuals show abnormal behaviours or behaviours outside of a normal context in a repeated and sustained manner. These presentations have previously been labelled obsessive compulsive, compulsive, stereotypic behaviour or stereotypies [63]. To avoid bracketing these behaviours with human obsessive compulsive disorder without evidence of a common aetiology, the authors will refer to this collection of behaviours as compulsive behaviours; described by Overall and Dunham [64] as “repetitive, ritualistic behaviours in excess of any required for normal function”. Differentials for compulsive behaviour traditionally leaned heavily towards neurological disorders and seizure activity and now include, but are not limited to central lesions, sensory neuropathies, dermatology, musculoskeletal, emotional conflict, conditioned behaviour [63]. In the authors’ experience there are often multiple contributing factors spanning emotional health, physiological health and cognitive processes.

Primary care veterinary clinicians are well placed to identify many of the clinical signs of excessive arousal. However, in the authors’ experience of general practice and referral practice, frequently core teaching in the veterinary curriculum does not equip clinicians to assess emotional health as a causal factor.

## 6. Behavioural Medicine Assessment

The standard diagnostic and treatment approach for each case assessed by Behavioural Referrals Veterinary Practice is summarised in Table 1. The process closely follows Maddison’s “Logical approach to clinical problem solving” [65], appropriate for medicine case work-up in all fields. From a behavioural medicine perspective, the diagnostic process is most closely aligned with the psychobiological approach [66] with an emphasis on optimising emotional, cognitive and physiological health to resolve presenting problems.

Cases may be put forward for discussion at multi-disciplinary rounds (MDR). These rounds were initiated by Professor Clare Rusbridge FRCVS and Dr Sarah Heath FRCVS to provide a service to their respective residents to discuss clinical cases from both a neurological and behavioural medicine perspective. They were so successful that they have since expanded to include many other veterinary disciplines. The authors attend these monthly, online meetings of specialists, residents and experienced clinicians from a variety of fields (including EBVS^®^ boarded veterinary specialists in Behavioural Medicine, Neurology, Anaesthesia and Analgesia, Veterinary and Comparative Nutrition, residents of those specialties, Animal Welfare residents and an ACPAT (Association of Charted Physiotherapists in Animal Therapy) registered physiotherapist currently seeing both human and non-human animal cases). Cases are presented and discussed with the influence of a variety of systems being taken into account. Suggestions for further investigations or treatment options are debated and, where appropriate, presented back to the caregiver.

## 7. Exemplary Case

A 7 month old male, entire, working-line Cocker Spaniel was presented with “fly catching behaviour” and “severe hallucination type behaviour”. The behavioural medicine consultation was a remote (video) consultation with the primary caregiver and the three household dogs present. Appendix A footage was provided by the caregivers both before the diagnostic consultation and as part of the structured follow up remote support. The patient lived in a household with a 13 year old male, neutered working Cocker Spaniel, a 10 year old male, neutered rescue Crossbreed, two adult caregivers and their two late-teenage daughters. The patient had been brought into the household at 7.5 weeks old and was described as the “perfect puppy” until the problem behaviours presented, though it was noted he had “always been intense” on walks. The in-home compulsive behaviours presented with sudden onset at 3.5 months of age with fly-snapping the first behaviour noted.

Work up by the primary care veterinarian and a neurology referral service took place within one week of the fly-snapping behaviour emerging, prior to referral to the authors’ behavioural medicine service. Work up included bloodwork, bile acid stimulation test, screening for toxoplasma, neospora and angiostrongylosus and brain magnetic resonance imaging, all of which were unremarkable. The patient was started on levetiracetam, 23 mg/kg TID, increased to 45 mg/kg TID for possible focal seizures and Nutracalm, 2 capsules SID with no notable effect reported by the caregivers. The patient remained on levetiracetam 45 mg/kg TID from 3.75 months old to the behavioural medicine consultation at 7 months old. Nutracalm was discontinued after two months with no deterioration noted.

The caregivers provided footage of problem behaviours occurring in a variety of contexts, see Appendix A. In Appendix A; the patient is sitting, with active shuffling and a high degree of muscular tone, intense focus towards multiple points on the bedding in front of the patient, ears are cocked forwards and tail, when visible, raised just above the level of the spine, jaws making intermittent snapping movements in the air between the patient’s nose and the bedding. In Appendix A; the patient is on a walk, standing in/pouncing through undergrowth, high degree of muscular tension, intense focus towards multiple points in the undergrowth with occasional moments of orientation towards the caregiver, ears are intermittently cocked forwards and held forwards but against the head with tail raised above the level of the spine and wagging rapidly, mouth is shut throughout. The patient also showed “manic” behaviours in the form of persistent, rapid pacing in the home and floor licking. For the remainder of this report, the authors will refer to this collection of behaviours in this case as “compulsive behaviours”. The behaviours were sufficiently frequent (see time budget below) and intense for the caregivers to be considering euthanasia on welfare grounds at the point of the diagnostic consultation.

Assessment of caregiver video footage along with specific history taking questions on dog-dog interactions within the household led to the conclusion that the patient’s relationship with the two other family dogs was within normal limits and the behaviour of the two other dogs did not seem to influence the presentation of the compulsive behaviours. The four caregivers had differing opinions on the management of the patient’s compulsive behaviours and this led to inconsistencies at times in the responses he received when demonstrating the behaviour. The primary caregiver was the only family member consistently present through the day with the rest of the family’s interactions relevant for short periods in the evening and at weekends. The primary caregiver reported a variety of actions she had taken in response to the patient showing compulsive behaviours. Initially she offered active distraction (cueing the dog to other activities or engaging in play which he enjoyed), followed by passive distraction (doing something that would interest the dog, without directing attention to the dog, which he responded to only whilst the activity persisted) but in more recent weeks had started to cue him to his crate (a cue he followed readily, settling if shut in) in response to his compulsive behaviours. Please note this caregiver lived in the UK. Animal welfare regulations vary from country to country including stipulations on when the use of a crate is permitted. The primary caregiver reported other family members would offer no response (ignoring the behaviour, with no effect reported if the primary caregiver was present) or occasionally verbally reprimand (no effect reported). The compulsive behaviours were shown at a higher frequency and intensity in the presence of the primary (adult female) caregiver, but were evident to a lesser reported intensity and duration in the rare event the family were at home without the primary caregiver. During all compulsive behaviour periods, the patient showed body language (ears forwards and tail in line with or above the level of the spine) associated with engaging emotional bias.

Day-to-day, the patient was active if the primary caregiver was present (all but 2 h of most days), sleeping only if put into his crate in this context. His night-time sleep quality and duration seemed good but day-time sleep was of poor duration and quality, unless crated. When resting outside of his crate, the patient would typically chew intensely on a hard chew (such as an antler). At the point of the diagnostic behavioural medicine consultation, a typical 24 h day for the patient would include 8 h crated sleep whilst the caregivers were in bed, 2 h crated sleep in two sessions whilst the caregivers were out of bed, 2 h in the garden with the other family dogs, 1–2 h being walked alone, 30 min eating and 9.5 h out of his crate in the home. During the vast majority of the time out of his crate in the home, the compulsive behaviours were shown if the primary caregiver was present.

A diagnosis of excessive emotional arousal was made following the initial behavioural medicine assessment. Largely, the emotions responsible for this were the engaging emotional systems of desire-seeking (related to resource acquisition, object play, predatory behaviour and in this case particularly, interactions with the primary caregiver) and social-play (relating to positive interactions with other dogs in the household). A mild-moderate level of the protective emotion, generalised anxiety, was present. In this case, generalised anxiety was deemed to, in part, contribute to the intensity of the vigilance the patient demonstrated towards his primary caregiver. The overall emotional arousal was not only excessive in intensity when the compulsive behaviours were shown, but also in duration throughout the day. The resulting poor sleep duration exacerbated the problem behaviours due to the missed opportunities for regulation of emotional arousal.

Cognitive factors were relevant in the form of inadvertent reinforcement of compulsive behaviours in their early presentation by the primary caregiver (when using active distraction). This led to an expectation from the patient of interaction from the caregiver when these behaviours were performed. The caregiver offering no interaction during more recent compulsive behaviour episodes likely induced some frustration as this expectation was thwarted, leading to a further increase in emotional arousal at these times. Cueing the patient to his crate consistently resulted in him readily moving there, seeming comfortable to be shut in and settling quickly. The crate appeared to be an effective discriminating stimulus by which the patient’s expectations of interaction or activity were managed and he would readily sleep on the majority of occasions he was confined to this location.

The influence of physical health factors was considered, but felt not to the be main or sole cause for the presenting problem behaviours. Analysis of video footage collected specifically for gait assessment showed poor hip extension bilaterally at run and simultaneous placement of the hind feet (“bunny hopping”) in the same gait. The role of potential underlying orthopaedic discomfort in particular was therefore discussed with the caregivers. In the author’s clinical experience a link between chronic pain and compulsive behavioural patterns is common and it is possible that investigation of this would be beneficial. An analgesia trial was declined by the caregivers.

The initial treatment plan is described in detail in Table 2. In brief, the main aim of treatment was to reduce emotional arousal by moderating desire-seeking responses, achieving appropriate day-time sleep, reducing frustration outbursts and facilitating appropriate independent decision making in the presence of the primary caregiver.

At one month review, the patient was reported to be showing no fly-snapping behaviour. He was frequently sleeping between thirty minutes and three hours each time he was cued to his crate in the day. The caregivers chose to start Skullcap and Valerian nutritional supplementation. A secondary treatment plan intervention of a no-attention cue [67] was taught to the caregivers to enable them to make it clear to the patient when he should expect no interaction from them, even when present in the same room, enabling the patient to reduce his vigilance enough to engage with independent activities or sleep without being cued to his crate. No attention cues involve a clear cue from the caregiver which predicts they will not interact (verbally, physically or with eye contact) with the dog. The author’s preferred cue is a visual cue of an object, meaning it can be presented and maintained whilst the caregivers perform other (non-dog related) activities. Initial introduction of the cue involves the caregiver offering calm but obvious interaction to the dog, then placing the object in sight and ignoring the dog momentarily before hiding the object and immediately giving the dog calm attention again. Duration of object-in-view time should be kept short initially, especially for patients very used to frequent caregiver interactions and with poor frustration tolerance. Gradually, over repeated sessions, the duration of object-in-view time (and therefore time the caregivers do not interact with the dog) can be increased. Typically, as the patient is seen to understand this cue, it is noted that they respond to the object by taking themselves off to perform independent activities, often including rest or sleep.

Three months after the initial behavioural medicine consultation, the caregivers reported the patient was able to take himself to his crate to sleep independently during the day, so cueing to the crate had reduced. He was also capable of sleeping or resting out of the crate with the other dogs with and without the primary caregiver present. A typical 24 h day for the patient at this point included 8 h crated sleep whilst the caregivers were in bed, estimate of 4–5 h sleep in multiple sessions, instigated primarily by the patient whilst the caregivers were out of bed, 2 h in the garden with the other family dogs, 1–2 h being walked alone, 30 min eating and 7.5 h out of his crate in the home. During the time out of his crate in the home, the patient was largely settled. He was signed off from the behavioural medicine service at this point.

Fourteen months after the initial behavioural medicine consultation, the authors requested an update with a view to reporting the case. The patient was still being managed as recommended and was reported by the caregivers to be “doing so well”, with the caregivers reporting no compulsive or other problem behaviours at this stage.

## 8. Case Discussion

The assessment and treatment of emotional health was an essential requirement in the successful management of this case. In particular, it is worth noting that whilst this patient presented with mild activation of protective emotional systems, he did not show a predominantly protective emotional bias. Patients with a protective bias show a majority of behaviours associated with activation of fear, anxiety, frustration, panic-grief or pain emotional systems and this is typically the focus of problem-behaviour assessment. This case demonstrates that emotional valence is not the only area which influences emotional and physical health. Total (engaging plus protective) emotional arousal should also be assessed in line with the Heath Model of Emotional Health [15,16] in order to establish if this is a contributing factor to the physical health condition that the patient presents with.

Assessment of temperament and impulsivity was not formally performed with validated methods in this patient. Formal assessment was not deemed likely to influence the treatment plan, however, use of tools such as the PANAS and DIAS questionnaires would have facilitated more accurate and objective progress tracking.

The reduction of emotional arousal in this case was largely achieved via facilitation of sleep opportunities for the patient. The patient was already partly motivated to sleep, doing so readily when cued to his crate and the door shut. However, this context needed to be provided for him to prioritise sleeping over other activities. It is important to note that Skullcap and Valerian supplementation could in part be responsible for the maintenance of the new-normal sleep routine for the patient. This supplementation was not started until some weeks after the initial treatment plan was outlined and a significant reduction in compulsive behaviours had been established prior to this point. Age-related development may also influence the maintenance of behaviours. Between initial behavioural medicine assessment and final follow up the patient experienced important developmental periods such as adolescence and development into adulthood which are acknowledged to significantly influence behavioural responses [68]. Such influences have not been specifically assessed in this case however the caregivers did not report associated behavioural changes.

Prior to behavioural medicine assessment, this patient was sleep-deprived. The author presented this case at MDR and the consequences of sleep deprivation were discussed. Behavioural medicine clinicians reported their experiences of poor emotional arousal regulation in cases with poor sleep duration. Neurology clinicians discussed the reports of hallucinations with sleep deprivation in the human literature [47,69] and questioned if this could be the mechanism motivating the presenting behaviour in this patient. Waters et al. [70] discuss in their review article of human sleep deprivation research that visual phenomena are the most common presenting sign in the early stage of sleep deprivation and in particular describe humans seeing “flashes, lights, dots” at this point. If we consider the patient in the case presented as experiencing sleep deprivation, in addition to having a very active engaging emotional system of desire-seeking, it is reasonable to theorise the “fly-snapping” behaviour may be targeted towards perceived visual stimuli. This motivation is completely unexplored and preliminary studies would need to be carefully designed given the difficulties of assessing hallucinations in non-human animals.

## 9. Conclusions

In practical terms, a thorough emotional health assessment is outside the scope of most primary care veterinary services. A brief assessment of sleep, reported problem behaviours and interactions with the caregiver can be made alongside the use of validated questionnaires pertinent to a case. This should be prioritised in cases presenting with physiological health conditions known, or likely, to have an underlying emotional arousal component; in particular chronic or intermittent gastrointestinal tract inflammation [71] and dermatological disease [12], feline lower urinary tract disease [1,2] and pain [7]. A more comprehensive emotional health assessment is indicated if caregivers report problem behaviours or if observed interactions with the caregiver involve highly intense interactions or behaviour motivated by fear and anxiety. Such an assessment is also necessary if patients achieve less than 16 h sleep a day at home or do not show signs of REM sleep [19].

The case example, offers a strong argument that a full veterinary behavioural medicine assessment of emotional and cognitive health should be considered part of a thorough approach to patients presenting with neurological disorders, especially those which are chronic, unresolving or unresponsive to medication. Furthermore, this case, along with the cited literature demonstrates why emotional health assessment should routinely be considered with any chronic physiological condition. As such, emotional health should be considered an essential part of the veterinary undergraduate curriculum and competency in emotional health assessment a necessary day one skill.

## Figures and Tables

**Table 1 animals-13-00465-t001:** A guide to the steps involved in a behavioural medicine case work-up.

Consultation Step	Description
Define the problem	As reported by the caregiver.
Refine the problem	Identify all signs within the patient’s day to day life which indicate emotional, physiological or cognitive health concerns.
Consider differentials	Identify emotional differentials: Framework: Panksepp’s affective systems [14]—including both negative/protective emotional systems (fear, anxiety, frustration, panic-grief and pain) and positive/engaging emotional systems (desire-seeking, social play, care and lust). Identify physiological differentials: Including any physical condition which impacts behaviour due to physical needs or limitations or via influence over emotional heath. Identify cognitive factors: Past learning events and what an individual therefore predicts regarding cause and effect. Cognitive health assessment regarding age or disease related cognitive decline.
Caregiver education	An essential step in most cases to motivate a caregiver to action the treatment plan. The emotional sink analogy from the Heath Model of Emotional Health [15] is a key tool for Behavioural Referrals Veterinary Practice clinicians to help caregivers understand and manage the patient’s emotional arousal and emotional valence (engaging compared to protective) bias.
Initial treatment plan	Framework: the Heath Model of Emotional Health [15,16] Designed to ensure activation of each emotional system is proportionate to the environment the patient is in, that the environment the patient is in is appropriate for their species specific needs and that any cognitive or physical health concerns are resolved or managed. Manipulate a patient’s total emotional arousal (engaging plus protective) such that it is not excessive in duration or intensity. Ensure sufficient engaging emotional activation is present and can be responded to appropriately and successfully to maintain good quality of life. Remove unnecessary protective emotions whilst ensuring justified protective emotions are proportional to environmental/social stimuli and the patient is able to successfully respond to them. Treatment may include environmental modifications, routine modification, changes to caregiver-patient interactions, management of inter-animal interactions, pheromone therapy, nutritional supplements and prescription medications.
Secondary treatment plan	Once the patient has responded sufficiently to the initial treatment plan to be showing appropriate emotional motivation(s) and associated behavioural responses to their environment, or sufficiently improved responses such that learning of positive associations is possible, a secondary treatment plan can be implemented. The secondary treatment plan may employ desensitisation and counter-conditioning or conditioned emotional response work. A range of cognitive training techniques to teach desirable behaviours in particular situations are considered, should these be useful for the patient and their caregivers. Often, once emotional responses are appropriate, patients begin to make good independent decisions with minimal or no caregiver intervention. When this occurs, the secondary treatment plan may become unnecessary, or adapted to a new set of aims.

**Table 2 animals-13-00465-t002:** Initial treatment plan, instigated following the diagnostic consultation.

Intervention	Description
Routine modification	Instigating “prophylactic”, routine time spent in the crate to ensure adequate opportunities for day-time sleep (until such a stage when the patient could sleep outside of the crate).
Changes to caregiver interactions	Caregiver education regarding management of emotional input to avoid intense arousal. This included moderating play to shorten bouts of high arousal play and introducing additional activities which interested but did not arouse the patient to the same degree, such as scent work. Ensuring clear “end” cues were offered to manage expectations at the end of interactive play. The end cue is a verbal cue the caregiver can offer to communicate play will cease. It is used to manage expectations regarding play availability and therefore to reduce frustration when play bouts with a caregiver end. It was taught in this case via the caregiver using their end cue, immediately ceasing all interaction for a moment before re-instigating a new game before the patient showed unwanted behaviours. The pause between the end cue and re-instigating play could be gradually increased until the patient was able to dis-engage from the caregiver following the end cue and perform appropriate, independent activities. In response to compulsive behaviours starting, all caregivers were to consistently cue the patient to his crate. Where possible, if compulsive behaviours could be predicted, the patient was to be cued to his crate just before these started. Changing affectionate interactions from human-style to be more dog-appropriate. Human-style affectionate interactions are typified by close physical, and often enveloping, contact and facial contact, each of which can signal uncertainty in dogs. It was recommended to show affection via physical contact including chin rubs, chest rubs, gentle strokes and ear fondles along with using structured play and training. Appropriate caregiver response to appeasement (the patient actively exchanging information with the caregivers at times of uncertainty). Caregivers frequently show human-style affection or reassurance in response to canine appeasement gestures. It was recommended to offer passive, calm interactions to the patient when he demonstrated appeasement, ensuring the patient could gather sufficient olfactory and tactile information from the caregiver to elicit all was well. This included offering a hand for the patient to sniff or allowing the patient to lean in contact with the caregiver whilst the caregiver demonstrated they felt calm and relaxed.
Activities to lower emotional arousal	Increased opportunities for independent licking and chewing. Increased opportunities for sleep (described in “routine modification).
Alter medication protocol	Slow wean of levetiracetam (in consultation with the neurology service the patient had seen). Anticipation of starting fluoxetine to manage arousal levels, however due to excellent response to the treatment plan listed above, this was not pursued.

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
