# Peer review of "Emotional Arousal Impacts Physical Health in Dogs: A Review of Factors Influencing Arousal, with Exemplary Case and Framework"

_animals, 2023, doi:10.3390/ani13030465_

Round 1

Reviewer 1 Report

Thank you for submitting this very interesting case report highlighting the importance of sleep, as well as assessing emotional health in a dog initially referred for a suspected neurological condition. This case is particularly valuable as it considers excessive positive emotional arousal as an important factor that needs addressing, which can often be overlooked with a tendency to focus on fear/anxiety, frustration, or the more generic “stress”

Some general comments:

1)      It would be clearer for the reader if you slightly altered the order of the information you present – i.e. in the first paragraph in section 2. Case example, you mention the age of the dog, then start with the description of the behaviours of concern. It is not until much later, and after you have already started to comment on the normality of certain interactions with other dogs being within normal limits, that you return to mention how old the dog was when acquired, the several lines later when the problem started. To aid the reader, a clearer timeline stated early in this section would be useful

2)      You describe this dog as being from working lines, with some insight into the daily activities. It would be useful to include a section on the dogs daily time budget (sleep, walks, crate time, play with owner, play/interactions with other dogs, engaging in chewing/feeding etc) before and after the consultation. Within this section, document the number of hours of sleep the dog obtained before and after, as this is a key feature of the case report, but details are lacking on this

Specific comments:

Ln 99 – can you please define “compulsive behaviours” and reference this definition – although labelling these behaviours as compulsive at the early stage in your report make it easier to use this term to describe them throughout, it does not lend itself well to considering other differentials/explanations for the behaviour, especially when you later suggest that there was clear inadvertent reinforcement.

Ln 99-101  -“The behaviours 99 were sufficiently frequent and intense for the caregivers to be considering euthanasia on 100 welfare grounds at the point of the diagnostic consultation.

It would be valuable to provide information on the frequency, and duration of these episodes – this is important when considering these as “compulsive” depending on which definition you use.

Ln 105-106 Can you provide further detail on the behaviour consultation process to clarify how the following was established – I assume it was either an in-clinic, or home visit assessment – stating this, plus how observations were made etc.  is suggested to qualify the statement “The patient’s interactions with the two other dogs were 105 within normal limits and the behaviour of the two other dogs did not seem to influence 106 the presentation of the compulsive behaviours.

Ln 107-110 – “The four caregivers had differing opinions 107 on the management of the patient’s compulsive behaviours and this lead to inconsisten-108 cies in the responses he received when demonstrating the behaviour (no response, passive 109 distraction, active distraction, verbal cues for alternative behaviours and being cued to his 110 crate).

Were all caregivers present at the consultation, or was this information collected in a different way, e.g. standardised questionnaire? Please clarify. Also, can you document what passive and active distraction comprised, and overall provide an account of whether any of these interventions were effective distractors, and/or whether they may have contributed to the maintenance of the problem?

Ln115-117 – Please include durations of treatment with levetiracetam and Nutracalm. Was there any data collected by the caregiver to monitor the response? i.e. was there any change in frequency or duration of episodes, even if not a resolution?

Ln 128-129 – can you please provide the initial data on sleep duration obtained which resulted in the assessment of quality and duration which “seemed good”.  This is particularly important, as in Ln 138-139, you go on to state “Poor sleep duration was 138 deemed to exacerbate the emotional arousal.”

Ln 136 – you refer earlier to the protective emotion of fear-anxiety, but here move to terminology of generalised anxiety. Can you please define this term, and how the assessment of generalised anxiety was made in this case as it is not clear from the rest of the assessment i.e. what was the evidence of this dog having fear-anxiety as a key emotion implicated in the problem behaviour? And based on what criteria was this anxiety deemed to be generalised? Vigilance in itself could relate to the engaging emotional systems you describe. You further go on to mention that generalised anxiety is “often functional in working dogs” – can you please expand on this further.

Ln 151-155 – “In particular, the role of potential un-151 derlying orthopaedic discomfort was discussed, following gait analysis. In the author’s 152 clinical experience a link between chronic pain and compulsive behavioural patterns is 153 common and it is therefore possible that investigation of this would be beneficial but an 154 analgesia trial was declined by the caregivers during the first phase of treatment.

Could you elaborate on what gait analysis revealed that resulted in this suspicion? It is not entirely clear what observations were made by the clinician.

Table 2 – typographical error in 2nd colum, line 4 “Jto”;

-          Additional information on what an “end cue” is and how it was taught, plus information of what other activities were used to minimise arousal should be included

-          Please clarify what you mean by “dog appropriate” and “appropriate caregiver response to appeasement”

Ln 165 - typographical error “valerian” rather than “valarian”.

Additionally, is there information on when skullcap/valerian was started in relation to when improvements were observed, and whether this was still being used to maintain the initial improvements reported at 15 months? Also, what dose was being used? As both of these herbs can potentially aid sleep, it is important to consider these may have been an effective part of the treatment, which should be included also in your discussion.

Ln 191 – I am not clear on what you mean by the phrase ”the patient was already partly motivated to sleep” – please explain.

Ln195 – you mention  multi-disciplinary rounds, however I cannot access the reference you provide. Could you please provide a short sentence on what these are/what disciplines were consulted within this process – you mention behaviour and neurology – were there others?  

Ln216-217 – please can you reference your guidance on sleep duration. 

Author Response

Thank you very much for your feedback. Please see my responses to each of your points below: 1) The first six paragraphs of section 2 have been re-ordered to clarify chronology. 2) Pre (lines 303-307) and post (lines 349-353) intervention time budget have been added. Ln 99 – define “compulsive behaviours” - this has now been expanded upon and defined (with reference) under the section “clinical signs of excessive or inappropriate emotional arousal” Ln 99-101 information on frequency of the compulsive behaviours has now been given within the time budget mentioned in point 2, with the time budget mentioned in line 303-307. Ln 105-106 the video format of the diagnostic consultation has been clarified (line 241) with clarification on the specific assessment of in-house dog-dog interactions in lines 270-273. Ln 107-110 these points have been included by expanding the paragraph commencing line 270. Ln 115 – this information on levetiracetam and Nutracalm has been included (lines 252-255). Ln 128-129 information on sleep duration is now provided in clearer form in the time budget mentioned in point 2. For clarity the “seemed good” refers to sleep whilst the caregivers are in bed and “poor sleep duration” referred to sleep whilst the caregivers are out of bed, which should now be clear in the article. Ln 136 these points have now been answered in the paragraph commencing line 145. I would also like to respond directly to your point “vigilance in itself could relate the engaging emotional systems”, I completely agree and feel in this case desire-seeking was very relevant. I have further clarified that the generalised anxiety described was only in part contributing to the vigilance and in particular, the intensity of the vigilance. Ln 151-155 detail of the gait assessment and abnormalities observed has been added in lines 328-329. Table 2 - Typographical error amended, thank you - A foot note has been added regarding the end cue. An example of low arousal activity has been provided in the table text. - Footnotes have been added to expand on “dog appropriate affection” and “appropriate response to appeasement” Ln 165 typo corrected, thank you. Comments on the potential influence of skullcap and valerian have been added to the discussion (lines 376-378). Ln 191 – explanation added for “partly motivated to sleep” (line 207) Ln 195 – A footnote describing MDR and the represented fields has been added and the reference availability followed up Ln 216-217 Reference added

Reviewer 2 Report

Case descriptions are always interesting but in this case, I think that a better explanation of the behaviour signs and an explanation of the diagnostic and therapeutic model used would help readers that are not familiar with the authors' approach. The discussion would benefit from a more in-depth analysis of the possible contributors to the dog's emotional and behaviour problems. 

Comments to the article ‘Assessment and treatment of emotional health is an essential 2 component of the treatment of complex physiological disor-3 ders; a case example and framework.’

Keywords: please add ‘dog’ to these keywords

Line 49

The definition 'protective emotional responses’ deserves a brief explanation about why and how the so called 'negative emotions' are protective and what this means from the animal’s point of view.

Table 2, page 3 – ‘Initial treatment plan’

‘Remove unnecessary protective emotions…’: excessive emotions?

Lines 99-100

Can you specify frequency and duration as reported by the caregivers?

Line 166

Please explain what a no-attention cue is and how it was introduced and taught

Line 176-179

This follow-up information is quite vague, were the compulsive behaviours reduced in duration and frequency or did they disappear?

Lines 181-185

This can be questionable. Inconsistent human animal interactions seemed to play a huge role and the main intervention was about giving this dog predictability in terms of routine and interactions. The owners’ inconsistency and unpredictability of interactions might have triggered frustration and anxiety (anticipation of punishment?).

Lines 218-end

I think it would be interesting to discuss about how the behavioural expressions of apparently positive emotions (engaging) and arousal can also be signs that the animal is not emotionally healthy and an excess of arousal and 'desire', when not met, can then trigger non healthy copying strategies - compulsive behaviours or displacement activities – that, in the ’sink’ model, is a way to 'discharge' a sink that become too full. Frustration can be triggered by an excess of seeking in impulsive individuals and impulsivity can also be a temperamental trait and the role of temperament in the onset of excessive/impulsive behaviour was not mentioned at all here. The discussion should have also included reflections about the possible developmental aspect of the disorder. The dog was ‘signed off from the behavioural medicine service’ at ten months, then the update requested by the authors was after fifteen months: important developmental periods like adolescence, juvenile and passage to adulthood and their influence of dog’s behaviour were not mentioned at all in the discussion.

Author Response

Thank you very much for your thorough and constructive feedback. Please see my response to your specific points below. Explanation of the behaviour signs is now included in paragraphs commencing lines 240 and 257, with more detail also added on the chronology of the presenting signs in paragraph commencing line 240. Please let me know if further detail is required and the nature of that detail. An explanation of the Heath Model in relation to the psychobiological approach has been included. The specific therapeutic model is described in table 1 “A guide to the steps involved in a behavioural medicine case work-up” and further detail has been added on how this particular consultation was conducted in lines 241-243 (regarding initial diagnostic consultation), 270-273 regarding assessment of in house dog-dog interactions and 328-329 regarding gait analysis. Further analysis on contributing factors has been added to the discussion, detailed in relation to your specific points below. Keywords: dog has been added Line 49 – a footnote explaining “protective emotional response” terminology has been added Table 2, page 3 “Initial treatment plan” – presume this references table 1, line 3. Initial treatment plan has been left unaltered. We discuss the manipulation of total emotional arousal (which correlates with your query of “excessive emotions”) in the first line of the third paragraph of this section and agree it is a key feature of this approach. However, the intent is to allow engaging emotional activation, so long as it is not detrimental to the patient’s welfare. Therefore, the line “remove unnecessary protective emotions” is intended to specifically refer to protective emotional activation only. The choice of the wording “unnecessary protective emotions” compared to “excessive protective emotions” is preferred as it implicitly includes contexts where protective emotions are triggered when none should be (such as an individual being fearful of safe novel objects including a new bed) in addition to contexts where some protective emotion is appropriate, but the individual shows an exaggerated response (such as excessive fear of traffic). Lines 99-100 Frequency and duration now covered as part of the time budget discussed lines 303-307. Line 166 A footnote describing the introduction of a no-attention cue has been added. Lines 176-179 clarity on compulsive behaviour expression has been added (now lines 246, 249, 257-267, paragraph starting 270). Lines 181-185 Thank you for this comment. I agree that typically, inconsistent human-animal interactions, especially those were punitive methods are employed induce fear-anxiety. It was not clear in the original manuscript the (small) influence of the inconsistencies between caregivers and I have now clarified this (line 274). In addition, I have commented on both the protective and engaging emotional activation in this section too (now lines 362-3). Lines 218-end thank you for this comment. An extra paragraph discussing excessive positive emotional arousal, along with temperamental testing has been included (commencing line 93) with addition of references 26-28. Mention of this has been included in closing conclusions (lines 370-372). Discussion of the potential influence of developmental periods has been included (lines 379-383). 

Reviewer 3 Report

I must admit that I am at least of three minds concerning this article. On the positive side I feel it very important to demonstrate that dog welfare can be tremendously impaired not only by NEGATIVE emotionality, both in our teaching and our consultancy we have similar experiences. But: I am not sure whether "Animals" is the best venue for this. Jouyyrnals like frontVetSci, BMC Vet Sci, JAVMA, J vet Behav might be better suited. And: The aurthors should decide whether they want it published as a case study or a review based on an exemplary case. In the first case they should shorten it consederably ( 3 - 4 pp at most) and leave out a lot e.g. the whole part on how to proceed in diagnostics & treatment in general. And this definitely would then be better suited in a vet/vet Behav journal.

Or, and then it might be suited for "Animals" after thorough rewriting, they focus on problems of activity budget, sleep/wake cycles etc.

In that case I have (sorry about that!!) some suggestions:

- in my opinion the case has more relevancy for the areas of time/activity budget, sleep patterns etc, the question of emotionality seems to be secondary. Thus I suggest to alter the title already, and include literature in these topics. I have asked a colleague in my team who just finished a book chapter (in German) to provide me with her literature list. As I am unable to upload that file here I am going to send it separately to the editor's office

- in any case please separate clearly the concepts of stress vs emiotonality. Stress, according to Donald Broom (eg 2001) is a condition of overcharging the adaptive systems with negative mid/long-term consequences for health and fitness, not an emotion. Emotions: We cannot be sure that everybody is familiar with e.g. Panksepp's 7 emotional systems. Thus please explain briefly what they are, and at least for those you consider relevant for your argument provide some info on neuroanatomy/-chemistry, also in relation to problems of addictive behaviour

-OCD/stereotypies: there is a lot of literature in this eg in the work of both Karen Overall's and Gerorgina Mason's(incl R Clubb) groups. These should be cited and included in the discussion

- welfare concepts (see Rooney/Bradshaw 2014 for a review, there also find Fraser, the 5 freedoms etc) should be referred to

- the so-called "Heath model" seems to be not a widely known concept, and the references to it also seem to be from marginal journals,. So please either explain that in detail, or leave it out and refer to more widely-known concepts instead (this is my concern in tjhe scores above on self-marketing)

- finally, a remark on crates: At least in the German speaking countries it is against animal welfare regulations to put a dog into a closed crate EXCEPT for transport or specific vet. treatment (with apprpriate monitoring and documentation by the vet). This should be mentioned, maybe there are similar regulations in other countries as well. at least a footnote requiring to consider country-specific regulations should be included.

Once again sorry for being so critical, but I think the impact of these important arguments made by the authors should be strengthened!!

Author Response

Thank you for your thorough and honest thoughts on the article. Your critique is appreciated. We have considered the two main directions you outlined for the manuscript and have opted to present a review based on an exemplary case. As such, adaptions have been made as suggested – outlined in relation to your specific points, below. The manuscript has been re-structured to clearly show the review sections, with the exemplary case following on. The review sections do include emotional arousal as the authors (supported by comments from the other reviewers) believe the valence of excessive emotional arousal to be an important aspect of this manuscript. Other review sections, as suggested, include the role of sleep and assessment via time budgets and clinical signs of excessive arousal (including compulsive behaviours). Whilst we agree with your distinction between stress and emotional arousal, we have elected not to discuss this further in this article. In the article, the term “stress” is only chosen once by the authors - in the key words - as we feel that people searching for this term may find the article interesting and following that used only twice, each time in quotations from other papers, with the reference for each cited. We feel involving the term stress in an article otherwise solely referring to emotional arousal and emotional valence would add unnecessary contradictory terminology for the reader. Panksepp’s emotional systems, and variations on this terminology have been discussed in the “emotional arousal” section. Affective neuroscience, an excellent and complete resource regarding the neuroanatomy and neurochemistry is referenced, and some aspects of this discussed in this section. The authors feel it is outside the scope of the manuscript to include full details on the affective systems’ neuroanatomy and neurochemistry. Overall and Mason’s publications on OCD/stereotypies have been reviewed along with further literature. A section on this is included in the “clinical signs of excessive or inappropriate emotional arousal” section along with references. Thank you for the recommendation to include Rooney and Bradshaw’s (2014) contribution to welfare concepts along with literature on the five freedoms. The authors have considered these important contributions and feel the new structure of the manuscript does not lend itself to inclusion of a separate section on welfare concepts. Animal health and welfare is a clear theme running through the article. The explanation for the Heath Model has been expanded with further referencing within the introductory paragraph to the Behavioural medicine assessment section. Crates – thank you for this point. A footnote has been added where “crate” is first mentioned. Your reference list on sleep articles has been read, with several papers from that now cited in this manuscript, thank you for the additional suggestions.

Round 2

Reviewer 2 Report

Dear Authors, 

thank you for revising this article, now more clear and therefore more likely to be a valid contribution to the field. 

Please consider the minor revisions below:

1. Line 91. Please add reference here

2. Table 1. in the section 'Caregiver education' a note about what  'emotional valence bias' is will be useful for non-specialistic readers. In the section 'Secondary treatment plan' the last sentence is more approriate in the discussion part.

3. Lines 380-382 (dog development) would probably deserve a reference. 

Author Response

Thank you for your further suggestions, please see responses to each below.
Line 91, reference added.
Table 1, Caregiver Education: clarification comment added.
Table 1, Secondary Treatment Plan: last sentence. The authors feel this sentence is correctly placed, but have added a second part to clarify the meaning of the sentence and thus why it is placed in the table.
Lines 380-382 (development): reference added.

Reviewer 3 Report

In section 3, temperament & trait, I suggest that the work by Turcsan, Kubinyi & Miklosi (2011) or the review by  Mklosi et al 2014 (in Kaminski & Marshall-Pescini eds) are referred to. Turcsan et al were rather successful in applying the big five, or at least four of them, and also some breed-typical data. Similarly Starling et al 2013 in Behab Proc. on breed charct. of the shy/bold model

Author Response

Thank you for these additional recommendations. The suggested references and some associated publications have been considered and additional points and citations included in section 3.